# Task Discovery: Finding the Tasks that Neural Networks Generalize on

**Andrei Atanov   Andrei Filatov   Teresa Yeo   Ajay Sohmshetty   Amir Zamir**

Swiss Federal Institute of Technology (EPFL)

`https://taskdiscovery.epfl.ch`

## Abstract

When developing deep learning models, we usually decide what task we want to solve then search for a model that generalizes well on the task. An intriguing question would be: *what if, instead of fixing the task and searching in the model space, we fix the model and search in the task space?* Can we find tasks that the model generalizes on? How do they look, or do they indicate anything? These are the questions we address in this paper.

We propose a *task discovery* framework that automatically finds examples of such tasks via optimizing a generalization-based quantity called *agreement score*. We demonstrate that one set of images can give rise to many tasks on which neural networks generalize well. These tasks are a reflection of the *inductive biases* of the learning framework and the *statistical patterns present in the data*, thus they can make a useful tool for analysing the neural networks and their biases. As an example, we show that the discovered tasks can be used to automatically create *"adversarial train-test splits"* which make a model fail at test time, *without changing the pixels or labels*, but by only selecting how the datapoints should be split between the train and test sets. We end with a discussion on human-interpretability of the discovered tasks.

## 1   Introduction

Deep learning models are found to generalize well, i.e., exhibit low test error when trained on human-labelled tasks. This can be seen as a consequence of the models' inductive biases that favor solutions with low test error over those which also have low training loss but higher test error. In this paper, we aim to find what are examples of other tasks[1] that are favored by neural networks, i.e., on which they generalize well. We will also discuss some of the consequences of such findings.

We start by defining a quantity called *agreement score* (AS) to measure how well a network generalizes on a task. It quantifies whether two networks trained on the same task with different training stochasticity (e.g., initialization) make similar predictions on new test images. Intuitively, a high AS can be seen as a necessary condition for generalization, as there cannot be generalization if the AS is low and networks converge to different solutions. On the other hand, if the AS is high, then there is a stable solution that different networks converge to, and, therefore, generalization is *possible* (see Appendix J). We show that the AS indeed makes for a useful metric and differentiates between human- and random-labelled tasks (see Fig. 1-center).

Given the AS as a prerequisite of generalization, we develop a *task discovery* framework that optimizes it and finds new tasks on which neural networks generalize well. Experimentally, we found that the same images can allow for many different tasks on which different network architectures generalize (see an example in Fig. 1-right).

---

[1]In the context of this paper, generally, a "task" is a labelling of a dataset, and any label set defines a "task".

36th Conference on Neural Information Processing Systems (NeurIPS 2022).

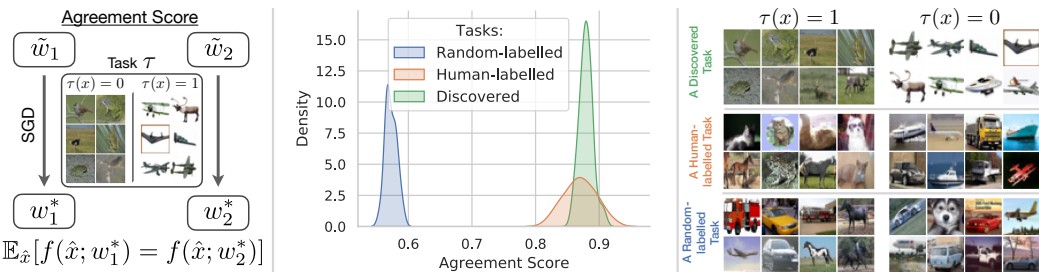

**Figure 1:** *Left:* The *agreement score* (AS) measures whether two networks trained on the task $\tau$ with different optimization stochasticity (e.g., initialization) make the same prediction on a test image $\hat{x}$. $f(\hat{x}, w)$ denotes a network with weights $w$ applied to an input $\hat{x}$. $\tilde{w}$ and $w^*$ denote initial and converged weights, respectively. *Center:* The agreement score successfully differentiates between human-labelled tasks based on CIFAR-10 original labels and random-labelled tasks. The *task discovery* framework finds novel tasks with high agreement scores. *Right:* Examples of a discovered, human-labelled (animals vs. non-animals) and random-labelled tasks on CIFAR-10. This particular discovered task looks visually distinct and seems to be based on the image background.

Finally, we discuss how the proposed framework can help us understand deep learning better, for example, by demonstrating its biases and failure modes. We use the discovered tasks to split a dataset into train test sets in an *adversarial* way instead of random splitting. After training on the train set, the network fails to make correct predictions on the test set. The adversarial splits can be seen as using the "spurious" correlations that exist in the datasets via discovered tasks. We conjecture that these tasks provide strong adversarial splits since task discovery finds tasks that are "favored" by the network the most. Unlike manually curated benchmarks that reveal similar failing behaviours [67, 47, 77], or pixel-level adversarial attacks [71, 40, 51], the proposed approach finds the adversarial split automatically and *does not need to change any pixels or labels* or collect new difficult or adversarial images.

## 2   Related Work

**Deep learning generalization.** It is not well understood yet why deep neural networks generalize well in practice while being overparameterized and having large complexity [26, 21, 59, 80, 75]. A line of work approaches this question by investigating the role of the different components of deep learning [1, 57, 2, 1, 53, 82, 72, 19] and developing novel complexity measures that could explain neural networks' generalization [58, 32, 46, 54, 69]. The proposed task discovery framework can serve as an experimental tool to shed more light on when and how deep learning models generalize. Also, compared to existing quantities for studying generalization, namely [81], which is based on the *training process only*, our agreement score is directly *based on test data and generalization*.

**Similarity measures between networks.** Measuring the similar between two networks is challenging and depends on a particular application. Two common approaches are to measure the similarity between hidden representations [45, 38, 41, 66, 44, 53] and the predictions made on unseen data [70]. Similar to our work, [19, 29, 64] also use an agreement score measure. In contrast to these works, which mainly use the AS as a metric, we turn it into an optimization objective to find new tasks with the desired property of good generalization.

**Bias-variance decomposition.** The AS used in this work can be seen as a measure of the variance term in the bias-variance decomposition of the test error [18, 4, 73, 14]. Recent works investigate how this term behaves in deep learning models [3, 55, 79, 56]. In contrast, we characterize its dependence on the task being learned instead of the model's complexity and find those that maximize the AS.

**Creating distribution shifts.** Distributions shifts can result from e.g. spurious correlations, under-sampling [77] or adversarial attacks [71, 40, 51]. To create such shifts to study the failure modes of networks, one needs to define them manually [67, 47]. We show that it is possible to *automatically* create *many* such shifts that lead to a significant accuracy drop on a given dataset using the discovered tasks (see Sec. 6 and Fig. 5-left). In contrast to other automatic methods to find such shifts, the proposed approach allows to creates many of them for a learning algorithm rather than for a single trained model [16], does not require additional prior knowledge [12] or adversarial optimization [42], and does not change pixel values [40] or labels.

**Data-centric analyses of learning.** Multiple works study how training data influences the final model. Common approaches are to measure the effect of removing, adding or mislabelling individual data points or a group of them [36, 37, 27, 28]. Instead of choosing which objects to train on, in task discovery, we choose how to label a fixed set of objects (i.e., tasks), s.t. a network generalizes well.

## 3 Agreement Score: Measuring Consistency of Labeling Unseen Data

In this section, we introduce the basis for the task discovery framework: the definition of a task and the agreement score, a measure of the task's generalizability. We then provide an empirical analysis of this score and demonstrate that it differentiates human- and random-labelled tasks.

**Notation and definitions.** Given a set of $N$ images $X = \{x_i\}_{i=1}^N$, $x \in \mathcal{X}$, we define a *task* as a binary labelling of this set $\tau : X \to \{0, 1\}$ (a multi-class extension is straightforward). We denote the corresponding labelled dataset as $\mathcal{D}(X, \tau) = \{(x, \tau(x))|x \in X\}$ and the set of all possible tasks (i.e., label sets) on $X$ as $\mathcal{T}_X$. We consider a learning algorithm $\mathcal{A}$ to be a neural network $f(\cdot; w) : \mathcal{X} \to [0, 1]$ with weights $w$ trained by SGD with cross-entropy loss. Due to the inherent stochasticity, such as random initialization and mini-batching, this learning algorithm induces a distribution over the weights given a dataset $w \sim \mathcal{A}(\mathcal{D}(X, \tau))$.

### 3.1 Agreement Score as a Measure of Generalization

A standard approach to measure the generalization of a learning algorithm $\mathcal{A}$ trained on $\mathcal{D}(X_{\mathrm{tr}}, \tau)$ is to measure the test error on $\mathcal{D}(X_{\mathrm{te}}, \tau)$. The test error can be decomposed into bias and variance terms [4, 73, 14], where, in our case, the stochasticity is due to $\mathcal{A}$ while the *train-test split is fixed*. We now examine how this decomposition depends on the task $\tau$. The bias term measures how much the average prediction deviates from $\tau$ and mostly depends on what are the test labels on $X_{\mathrm{te}}$. The variance term captures how predictions of different models agree with each other and does not depend on the task's test labels but only on training labels through $\mathcal{D}(X_{\mathrm{tr}}, \tau)$. We suggest measuring the generalizability of a task $\tau$ with an *agreement score*, as defined below.

For a given train-test split $X_{\mathrm{tr}}, X_{\mathrm{te}}$ and a task $\tau$, we define the agreement score (AS) as:

$$\mathrm{AS}(\tau; X_{\mathrm{tr}}, X_{\mathrm{te}}) = \mathbb{E}_{w_1, w_2 \sim \mathcal{A}(\mathcal{D}(X_{\mathrm{tr}}, \tau))} \mathbb{E}_{x \sim X_{\mathrm{te}}} [f(x; w_1) = f(x; w_2)], \tag{1}$$

where the first expectation is over different models trained on $\mathcal{D}(X_{\mathrm{tr}}, \tau)$ and the inner expectation is averaging over the test set. Practically, this corresponds to training two networks from different initializations on the training dataset labelled by $\tau$ and measuring the agreement between these two models' predictions on the test set (see Fig. 1-left and the inner-loop in Fig. 3-left).

The AS depends only on the task's training labels, thus, test labels are not required. However, it is tightly connected to the test error, as having a high-AS task $\tau$ that labels training data, one can construct a task with a high test accuracy. We note, however, that when the task is given and fixed, the high-AS provides only a necessary but not a sufficient condition for a high test accuracy as test labels can take any values in general (e.g., be adversarial as in Sec. 5), thus, it cannot be used to predict the test accuracy, in this case [29, 35]. Refer to Appendix J for a more in-depth discussion.

### 3.2 Agreement Score Behaviour for Random- and Human-Labelled Tasks

Here, we demonstrate that the AS exhibits the desired behaviour of differentiating human-labelled from random-labelled tasks. We take the set of images $X$ from the CIFAR-10 dataset [39] and split the original training set into 45K images for $X_{\mathrm{tr}}$ and 5K for $X_{\mathrm{te}}$. We split 10 original classes differently into two sets of 5 classes to construct 5-vs-5 binary classification tasks $\tau_{\mathrm{hl}}$. Out of all $\binom{10}{5}$ tasks, we randomly sub-sample 20 ones to form the set of *human-labelled tasks* $T_{\mathrm{HL}}$. We construct the set of 20 *random-labelled tasks* $T_{\mathrm{RL}}$ by generating binary labels for all images randomly and fixing them throughout the training, similar to [81]. We use ResNet-18 [24] architecture and Adam [34] optimizer as the learning algorithm $\mathcal{A}$, unless otherwise specified. We measure the AS by training two networks for 100 epochs, which is enough to achieve zero training error for all considered tasks.

**AS differentiates human- from random-labelled tasks.** Fig. 1-center shows that human-labelled tasks have a higher AS than random-labelled ones. This coincides with our intuition that one should not expect generalization on a random task, for which AS is close to the chance level of 0.5. Note, that the empirical distribution of the AS for random-labelled tasks (Fig. 2-left) is an unbiased estimation of the AS distribution over all possible tasks, as $T_{\mathrm{RL}}$ are uniform samples from $\mathcal{T}_{X_{\mathrm{tr}}}$. This suggests that high-AS tasks do not make for a large fraction of all tasks. A high AS of human-labelled tasks

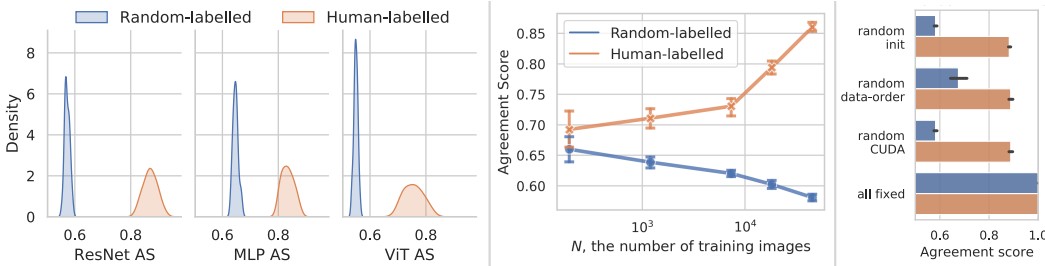

**Figure 2: Agreement score for human- and random-labelled tasks.** *Left:* AS measured on three architectures: ResNet-18, MLP and ViT. *Center:* AS measured on ResNet-18 for different numbers of training images $N$. The standard deviation is over four random tasks and three data splits. *Right:* Ablating the sources of stochasticity present in $\mathcal{A}$. Each row shows when one of the 1) initialization, 2) data-order or 3) CUDA is stochastic and the other two sources are fixed, and the bottom is when all the sources are fixed (see Sec. 3.2). The differentiation between human- and random-labelled tasks stably persists across different architectures, data sizes and sources of randomness.

is also consistent with the understanding that a network is able to generalize when trained on the original CIFAR-10 labels.

**How does the AS differ across architectures?** In addition to ResNet-18, we measure the AS using MLP [70] and ViT architectures [15] (see Appendix L.3). Fig. 2-left shows that the AS for all the architectures distinguishes tasks from $T_{\mathrm{RL}}$ and $T_{\mathrm{HL}}$ well. MLP and ViT have lower AS than ResNet-18, aligned with our understanding that convolutional networks should exhibit better generalization on this small dataset due to their architectural inductive biases. Similar to the architectural analysis, we provide the analysis on the data-dependency of the AS in Appendix C.

**How does the AS depend on the training size?** When only too little data is available for training, one could expect the stochasticity in $\mathcal{A}$ to play a bigger role than data, i.e., the agreement score decreases with less data. This intuition is consistent with empirical evidence for human-labelled tasks, as seen in Fig. 2-center. However, for random-labelled tasks, the AS increases with less data (but they still remain distinguishable from human-labelled tasks). One possible reason is that when the training dataset is very small, and the labels are random, the two networks do not deviate much from their initializations. This results in similar networks and consequently a high AS, but basically irrelevant to the data and uninformative.

**Any stochasticity is enough to provide differentiation.** We ablate the three sources of stochasticity in $\mathcal{A}$: 1) initialization, 2) data-order and 3) CUDA stochasticity [31]. The system is fully deterministic with all three sources fixed, thus, AS=1. Interestingly, when *any* of these variables change between two runs, the AS drops, creating separation between human- and random-labelled tasks.

These empirical observations show that the AS well differentiates between human- and random-labelled tasks across multiple architectures, dataset sizes and sources of stochasticity.

## 4 Task Discovery via Meta-Optimization of the Agreement Score

As the previous section shows, AS is a useful measure of network's generalization on a given task. A natural question then arises: *are there high-AS tasks other than human-labelled ones, and what are they?* In this section, we establish a *task discovery* framework to automatically search for these tasks and study this question computationally.

### 4.1 Task Space Parametrization and Agreement Score Meta-Optimization

Our goal is to find a task $\tau$ that maximizes $\mathrm{AS}(\tau)$. It is a high-dimensional discrete optimization problem which is computationally hard to solve. In order to make it differentiable and utilize more efficient first-order optimization methods, we, first, substitute the 0-1 loss in Eq. 1 with the cross-entropy loss $l_{\mathrm{ce}}$. Then, we parameterize the space of tasks with a *task network* $t_\theta : \mathcal{X} \to [0, 1]$ and treat the AS as a function of its parameters $\theta$. This basically means that we view the labelled training dataset $\mathcal{D}(X_{\mathrm{tr}}, \tau)$ and a network $t_\theta$ with the same labels on $X_{\mathrm{tr}}$ as being equivalent, as one can always train a network to fit the training dataset.

Given that all the components are now differentiable, we can calculate the gradient of the AS w.r.t. task parameters $\nabla_\theta AS(t_\theta)$ by unrolling the inner-loop optimization and use it for gradient-based

optimization over the task parameters $\theta$. This results in a meta-optimization problem where the inner-loop optimization is over parameters $w_1, w_2$ and the outer-loop optimization is over $\theta$ (see Fig. 3-left).

Evaluating the meta-gradient w.r.t. $\theta$ has high memory and computational costs, as we need to train two networks from scratch in the inner-loop. To make it feasible, we limit the number of inner-loop optimization steps to 50, which we found to be enough to separate the AS between random and human-labelled tasks and provide a useful learning signal (see Appendix L.1). We additionally use gradient checkpointing [8] after every inner-loop step to avoid linear memory consumption in the number of steps. This allows us to run the discovery process for the ResNet-18 model on the CIFAR-10 dataset using a single 40GB A100. See Appendix L.2 for more details.

### 4.2 Discovering *a Set* of Dissimilar Tasks with High Agreement Scores

The described AS meta-optimization results in only a single high-AS task, whereas there are potentially many of them. Therefore, we would like to discover a set of tasks $T = \{t_{\theta_1}, \ldots, t_{\theta_K}\}$. A naive approach to finding such a set would be to run the meta-optimization from $K$ different intializations of task parameters $\theta$. However, this results in a set of similar (or even the same) tasks, as we observed in the preliminary experiments. Therefore, we aim to discover *dissimilar* tasks to better represent the set of all high-AS tasks. We measure the similarity between two tasks on a set $X$ as follows:

$$\text{sim}_X(\tau_1, \tau_2) = \max \left\{ \mathbb{E}_{x \sim X}[\tau_1(x) = \tau_2(x)], \mathbb{E}_{x \sim X}[\tau_1(x) = 1 - \tau_2(x)] \right\}, \quad (2)$$

where the maximum accounts for the labels' flipping. Since this metric is not differentiable, we, instead, use a differentiable loss $L_{\text{sim}}$ to minimize the similarity (defined later in Sec. 4.3, Eq. 5). Finally, we formulate the task discovery framework as the following optimization problem over $T$:

$$\underset{T=\{t_{\theta_1}, \ldots, t_{\theta_K}\}}{\arg \max} \quad \mathbb{E}_{t_\theta \sim T} \text{AS}(t_\theta) - \lambda \cdot L_{\text{sim}}(T). \quad (3)$$

We show the influence of $\lambda$ on task discovery in Appendix H. Note that this naturally avoids discovering trivial solutions that are highly imbalanced (e.g., labelling all objects with class 0) due to the similarity loss, as these tasks are similar to each other, and a set $T$ that includes them will be penalized.

In practice, we could solve this optimization sequentially – i.e., first find $t_{\theta_1}$, freeze it and add it to $T$, then optimize for $t_{\theta_2}$, and so on. However, we found this to be slow. In the next Sec. 4.3, we provide a solution that is more efficient, i.e., can find more tasks with less compute.

***Regulated* task discovery.** The task discovery formulation above only concerns with finding high-AS tasks – which is the minimum requirement for having generalizable tasks. One can introduce additional constraints to *regulate* the discovery process, e.g., by adding a regularizer to Eq. 3 or via the task network's architectural biases. This approach can allow for a guided task discovery to favor the discovery of particular tasks. We provide an example of a regulated discovery in Sec. 5.1 by using self-supervised pre-trained embeddings as the input to the task network.

### 4.3 Modelling The Set of Tasks with an Embedding Space

Modelling every task by a separate network increases memory and computational costs. In order to amortize these costs, we adopt an approach popular in multi-task learning and model task networks with a shared encoder $e(\cdot; \theta_e) : \mathcal{X} \to \mathbb{R}^d$ and a task-specific linear head $\theta_l \in \mathbb{R}^d$, so that $t_\theta(x) = e(x; \theta_e)^\top \theta_l$. See Fig. 3-right for visualization. Then, instead of learning a fixed set of different task-specific heads, we aim to learn an embedding space where *any* linear hyperplane gives rise to a high-AS task. Thus, an encoder with parameters $\theta_e$ defines the following set of tasks:

$$T_{\theta_e} = \{t_\theta \mid \theta = (\theta_e, \theta_l), \theta_l \in \mathbb{R}^d\}. \quad (4)$$

This set is not infinite as it might seem at first since many linear layers will correspond to the same shattering of $X$. The size of $T_{\theta_e}$ as measured by the number of unique tasks on $X$ is only limited by the dimensionality of the embedding space $d$ and the encoder's expressivity. Potentially, it can be as big as the set of all shatterings when $d = |X| - 1$ and the encoder $e$ is flexible enough [74].

We adopt a uniformity loss over the embedding space [76] to induce dissimilarity between tasks:

$$L_{\text{sim}}(T_{\theta_e}) = L_{\text{unif}}(\theta_e) = \log \mathbb{E}_{x_1, x_2} \exp \left\{ \alpha \cdot \frac{e(x_1; \theta_e)^\top e(x_2; \theta_e)}{\|e(x_1; \theta_e)\| \cdot \|e(x_2; \theta_e)\|} \right\}, \quad (5)$$

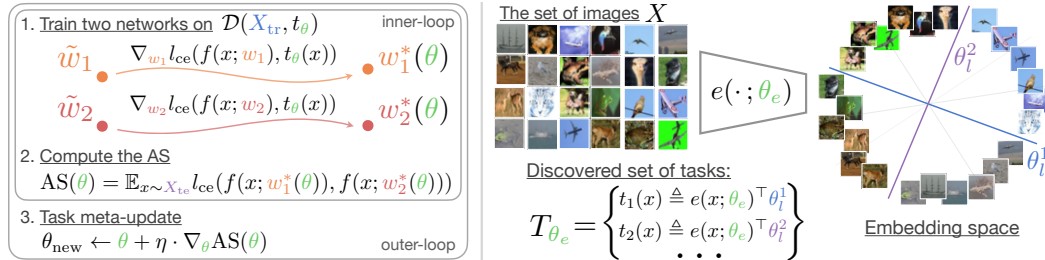

**Figure 3:** *Left:* Agreement score meta-optimization. *Inner-loop:* given the task $t_\theta \in T$, two networks $w_1, w_2$ optimize the cross-entropy loss $l_{ce}$ on the training set labelled by the task $t_\theta$ (Sec. 4.1). After training, the agreement between two networks is calculated on the test set. *Outer-loop:* the task parameters $\theta$ are updated with the meta-gradient of the AS. *Right:* the task-space paramertrization with the shared encoder (Sec. 4.3). The encoder $e(\cdot; \theta_2)$ (after projection) distributes images uniformly on the sphere in the embedding space. Different tasks are then formed by applying a linear classifier with weights, e.g., $\theta_l^1$ and $\theta_l^2$, passing through the origin. The corresponding set of tasks $T_{\theta_e}$ consists of *all* such linear classifiers in this space. This results in a more efficient task discovery framework than modelling each task with a separate network.

where the expectation is taken w.r.t. pairs of images randomly sampled from the training set. It favours the embeddings to be uniformly distributed on a sphere, and, if the encoder satisfies this property, any two orthogonal hyper-planes $\theta_l^1 \perp \theta_l^2$ (passing through origin) give rise to two tasks with the low similarity of 0.5 as measured by Eq. 2 (see Fig. 3-right).

In order to optimize the AS averaged over $T_{\theta_e}$ in Eq. 3, we can use a Monte-Carlo gradient estimator and sample one hyper-plane $\theta_l$ at each step, e.g., w.r.t. an isotropic Gaussian distribution, which, on average, results in dissimilar tasks given that the uniformity is high. As a result of running the task discovery framework, we find the encoder parameters $\theta_e^*$ that optimize the objective Eq. 3 and gives rise to the corresponding set of tasks $T_{\theta_e^*}$ (see Eq. 4 and Fig. 3-right).

Note that the framework outlined above can be straightforwardly extended to the case of discovering multi-way classification tasks as we show in Appendix D, where instead of sampling a hyperplane that divides the embedding space into two parts, we sample $K$ hyperplanes that divide the embedding space into $K$ regions and give rise to $K$ classes.

**Towards *the space* of high-AS tasks.** Instead of creating *a set* of tasks, one could seek to define *a space* of high-AS tasks. That is to define *a basis set* of tasks and a binary operation on a set of tasks that constructs a new task and preserves the AS. The proposed formulation with a shared embedding space can be seen as a step toward this direction. Indeed, in this case, the structure over $T_{\theta_e^*}$ is imposed implicitly by the space of liner hyperplanes $\theta_l \in \mathbb{R}^d$, each of which gives rise to a high-AS task.

## 4.4 Discovering High-AS Tasks on CIFAR-10 Dataset

In this section, we demonstrate that the proposed framework successfully finds dissimilar tasks with high AS. We consider the same setting as in Sec. 3.2. We use the encoder-based discovery described in Sec. 4.3 and use the ResNet-18 architecture for $e(\cdot; \theta_e)$ with a linear layer mapping to $\mathbb{R}^d$ (with $d = 32$) instead of the last classification layer. We use Adam as the meta-optimizer and SGD optimizer for the inner-loop optimization. Please refer to Appendix L.2 for more details.

We optimize Eq. 3 to find $\theta_e^*$ and sample 32 tasks from $T_{\theta_e^*}$ by taking $d$ orthogonal hyper-planes $\theta_l$. We refer to these 32 tasks as $T_{\text{ResNet}}$. The average similarity (Eq. 2) between all pairs of tasks from this set is 0.51, close to the smallest possible value of 0.5. For each discovered task, we evaluate its AS in the same way as in Sec. 3.2 (according to Eq. 1). Fig. 1-center demonstrates that the proposed task discovery framework successfully finds tasks with high AS. See more visualizations and analyses of these discovered tasks in Sec. 5 and Appendix B.

**Random networks give rise to high-AS tasks.** Interestingly, in our experiments, we found that if we initialize a task network randomly (standard uniform initialization from PyTorch [63]), the corresponding task (after applying softmax) will have a high AS, on par with human-labelled and discovered ones ($\approx 0.85$). Different random initializations, however, give rise to very similar tasks (e.g., 32 randomly sampled networks have an average similarity of 0.68 compared to 0.51 for the discovered tasks. See Appendix K for a more detailed comparison). Therefore, a naive approach of sampling random networks does not result in an efficient task discovery framework, as one needs to

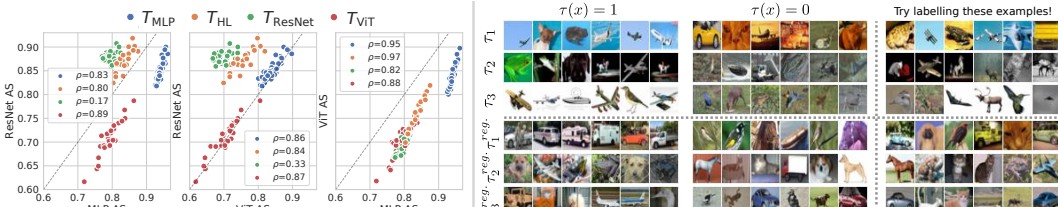

**Figure 4:** *Left:* The AS for tasks from $T_{\text{HL}}, T_{\text{ResNet}}, T_{\text{MLP}}, T_{\text{ViT}}$ measured on all three architectures ($x$ and $y$ axis). *Right:* Sample images for each class of our discovered tasks are shown in the first two columns. We show some unlabelled examples in the third column for the reader to guess the label. The answers are in the Appendix B. These images have been sampled to be the most discriminative, as measured by the network predicted probability. *Right* (top): Some of the discovered tasks from the unregulated version of task discovery seem to be based on color patterns, e.g. $\tau_1(x) = 1$ are images with similar blue color, which make sense as a learnable task and are expected (see Sec. 5.1). *Right* (bottom): The same, but for a regulated version where the encoder was pretrained with SimCLR. The tasks seems to correspond more to semantic tasks e.g. $\tau_1^{reg\cdot}(x) = 1$ are images of vehicles with different backgrounds. As the pretraining encourages the encoder to be invariant to color, it seems to be biased towards semantic information. Thus, the framework is able to pick up on the inductive biases from SimCLR pretraining, via the discovered tasks.

draw prohibitively many of them to get sufficiently dissimilar tasks. Note, that this result is specific to the initialization scheme used and not *any* instantiation of the network's weights necessarily results in a high-AS task.

# 5 Empirical Study of the Discovered Tasks

In this section, we first perform a qualitative analysis of the discovered tasks in Sec. 5.1. Second in Sec. 5.2, we discuss the human-interpretability of the discovered tasks and whether human-labelled tasks should be expected to be discovered. Then, we analyze how discovered tasks differ for three different architectures in Sec. 5.3. We also study the dependency of the AS on the test data domain and show the results in Appendix C in the interest of space.

## 5.1 Qualitative Analyses of the Discovered Tasks

Here we attempt to analyze the tasks discovered by the proposed discovery framework. Fig. 4-top-right shows examples of images for three sample discovered tasks found in Sec. 4.4. For ease of visualization, we selected the most discriminative images from each class as measured by the network's predicted probability. In the interest of space, the visuals for all tasks are in the Appendix B, alongside some post hoc interpretability analysis.

Some of these tasks seem to be based on color, e.g. the class 1 of $\tau_1$ includes images with blue (sky or water) color, and the class 0 includes images with orange color. Other tasks seem to pick up other cues. These are basically reflections of the *statistical patterns present in the data* and the *inductive biases of the learning architecture*.

**Regulated task discovery** described in Sec. 4.2 allows us to incorporate additional information to favor the discovery of specific tasks, e.g., ones based on more high-level concepts. As an example, we use self-supervised contrastive pre-training that learns embeddings invariant to the employed set of augmentations [20, 22, 5, 52]. Specifically, we use embeddings of the ResNet-50 [24] trained with SimCLR [9] as the input to the encoder $e$ instead of raw images, which in this case is a 2-layer fully-connected network. Note that the AS is still computed using the raw pixels.

Fig. 4-bottom-right shows the resulting tasks. Since the encoder is now more invariant to colour information due to the colour jittering augmentation employed during pre-training [9], the discovered tasks seem to be more aligned with CIFAR-10 classes; e.g. samples from $\tau_1^{reg}$'s class 1 show vehicles against different backgrounds. Note that task discovery regulated by contrastive pre-training only provides a tool to discover tasks invariant to the set of employed augmentations. The choice of augmentations, however, depends on the set of tasks one wants to discover. For example, one should not employ a rotation augmentation if they need to discover view-dependent tasks [78].

## 5.2 On Human-Interpretability and Human-Labelled Tasks

In this section, we discuss *i)* if the discovered tasks should be visually interpretable to humans, *ii)* if one should expect them to contain human-labelled tasks and if they do in practice, and *iii)* an example of how the discovery of human-labelled tasks can be promoted.

**Should the discovered high-AS tasks be human-interpretable?** In this work, by "human-interpretable tasks", we generally refer to those classifications that humans can visually understand or learn sufficiently conveniently. Human-labelled tasks are examples of such tasks. While we found in Sec. 3.2 that such interpretable tasks have a high AS, the opposite is not true, i.e., not *all* high-AS tasks should be visually interpretable by a human – as they reflect the inductive biases of the particular learning algorithm used, which are not necessarily aligned with human perception. The following "green pixel task" is an example of such a high-AS task that is learnable and generalizable in a statistical learning sense but not convenient for humans to learn visually.

**The *"green pixel task"*.** Consider the following simple task: the label for $x$ is 1 if the pixel $p$ at a fixed position has the green channel intensity above a certain threshold and 0 otherwise. The threshold is chosen to split images evenly into two classes. This simple task has the AS of approximately $0.98$, and a network trained to solve it has a high test accuracy of $0.96$. Moreover, the network indeed seems to make the predictions based on this rule rather than other cues: the accuracy remains almost the same when we set all pixels but $p$ to random noise and, on the flip side, drops to the chance level of 0.5 when we randomly sample $p$ and keep the rest of the image untouched. This suggests that the network captured the underlying rule for the task and generalizes well in the statistical learning sense. However, it would be hard for a human to infer this pattern by only looking at examples of images from both classes, and consequently, it would appear like an uninterpretable/random task to human eyes.

It is sensible that many of the discovered tasks belong to such a category. This indicates that finding more human-interpretable tasks would essentially require *bringing in additional constraints and biases* that the current neural network architectures do not include. We provide an example of such a promotion using the SSL regulated task discovery results below.

**Do discovered tasks contain human-labelled tasks?** We observe that the similarity between the discovered and most human-labelled tasks is relatively low (see Fig. 11-left and Appendix E for more details). As mentioned above, human-labelled tasks make up only a small subset of all tasks with a high AS. The task discovery framework aims to find different (i.e., dissimilar) tasks from this set and not necessarily *all* of them. In other words, there are many tasks with a high AS other than human-labelled ones, which the proposed discovery framework successfully finds.

As mentioned above, introducing additional inductive biases would be necessary to promote finding more human-labelled tasks. We demonstrate this by using the tasks discovered with the SimCLR pre-trained encoder (see Sec. 5.1). Fig. 11-right shows that the recall of human-labelled tasks among the discovered ones increases notably due to the inductive biases that SimCLR data augmentations bring in.

## 5.3 The Dependency of the Agreement Score and Discovered Tasks on the Architecture

In this section, we study how the AS of a task depends on the neural network architecture used for measuring the AS. We include human-labelled tasks as well as a set of tasks discovered using different architectures in this study. We consider the same architectures as in Sec. 3.2: ResNet-18, MLP, and ViT. We change both $f$ and $e$ to be one of MLP or ViT and run the same discovery process as in Sec. 4.4. As a result, we obtain three sets: $T_{\text{ResNet}}, T_{\text{MLP}}, T_{\text{ViT}}$, each with 32 tasks. For each task, we evaluate its AS on all three architectures. Fig. 4-left shows that high-AS tasks for one architecture do not necessary have similar AS when measured on another architecture (e.g., $T_{\text{ResNet}}$ on ViT). For MLP and ViT architectures, we find that ASs correlate well for all groups for tasks, which is aligned with the understanding that these architectures are more similar to each other than to ResNet-18.

More importantly, we note that comparing architectures on any single group of tasks is not enough. For example, if comparing the AS for ResNet-18 and MLP only on human-labelled tasks $T_{\text{HL}}$, one might conclude that they correlate well ($\rho = 0.8$), suggesting they generalize on similar tasks. However, when the set $T_{\text{ResNet}}$ is taken into account, the conclusion changes ($\rho = 0.17$). Thus, it is important to analyse the different architectures on a broader set of tasks not to bias our understanding, and the proposed task discovery framework allows for more complete analyses.

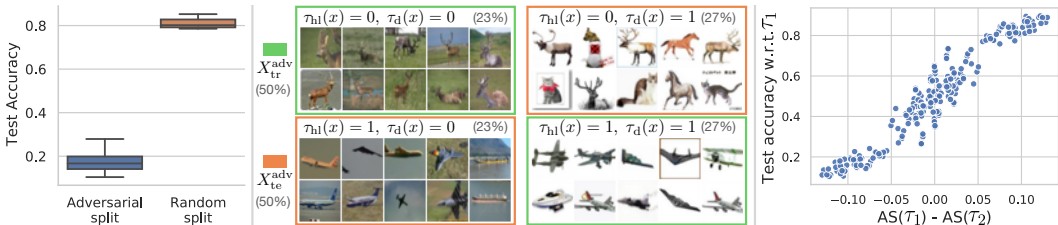

**Figure 5: Adversarial splits on CIFAR-10.** *Left:* Test accuracy for human-labelled tasks $\tau_{\mathrm{hl}}$ on random and adversarial splits. The boxplot distribution is over four $\tau_{\mathrm{hl}}$ tasks for the random split and 24 $(\tau_{\mathrm{hl}}, \tau_{\mathrm{d}})$ pairs for the adversarial one. Training with the adversarial split results in a significant test performance drop. *Center:* Example of an adversarial split, constructed such that images from train and test sets have the opposite correlation between $\tau_{\mathrm{hl}}$ and $\tau_{\mathrm{d}}$ (the percentage of images in each group is shown in brackets). The model seems to learn to make its predictions based on $\tau_{\mathrm{d}}$ instead of $\tau_{\mathrm{hl}}$ (hence, the significant test accuracy drop). *Right:* Each dot corresponds to a pair of tasks $(\tau_1, \tau_2)$, where the *y-axis* is the test accuracy after training on the corresponding adversarial split to predict $\tau_1$, and the *x-axis* is the difference in the AS of these two tasks. The plot suggests that the network *favors learning the task with a higher AS*: the higher the AS of $\tau_1$ is, the higher the accuracy w.r.t. $\tau_1$ is. Thus, for an adversarial split to be successful, the AS of the target task $\tau_1$ should be lower than the AS of $\tau_2$.

## 6 Adversarial Dataset Splits

In this section, we demonstrate how the discovered tasks can be used to reveal biases and failure modes of a given network architecture (and training pipeline). In particular, we introduce an *adversarial split* – which is a train-test dataset split such that the test accuracy of the yielded network significantly drops, compared to a random train-test split. This is because the AS can reveal what tasks a network *favors* to learn, and thus, the discovered tasks can be used to construct such adversarial splits.

### 6.1 Creating Adversarial Train-Test Dataset Partitions Using Discovered High-AS Tasks

For a given human-labelled task $\tau_{\mathrm{hl}}$, let us consider the standard procedure of training a network on $\mathcal{D}(X_{\mathrm{tr}}, \tau_{\mathrm{hl}})$ and testing it on $\mathcal{D}(X_{\mathrm{te}}, \tau_{\mathrm{hl}})$. The network usually achieves a high test accuracy when the dataset is split into train and test sets randomly. Using discovered tasks, we show how to construct *adversarial* splits on which the test accuracy drops significantly after training. To do that, we take a discovered task $\tau_{\mathrm{d}}$ with a high AS and construct the split, s.t. $\tau_{\mathrm{hl}}$ and $\tau_{\mathrm{d}}$ have the same labels on the training set $X_{\mathrm{tr}}^{\mathrm{adv}}$ and the opposite ones on the test set $X_{\mathrm{te}}^{\mathrm{adv}}$ (see Fig. 5-center):

$$X_{\mathrm{tr}}^{\mathrm{adv}} = \{x \,|\, \tau_{\mathrm{hl}}(x) = \tau_{\mathrm{d}}(x), \, x \in X\}, \quad X_{\mathrm{te}}^{\mathrm{adv}} = \{x \,|\, \tau_{\mathrm{hl}}(x) \neq \tau_{\mathrm{d}}(x), \, x \in X\}. \tag{6}$$

Fig. 5-left shows that for various pairs of $(\tau_{\mathrm{hl}}, \tau_{\mathrm{d}})$, the test accuracy on $\mathcal{D}(X_{\mathrm{te}}^{\mathrm{adv}}, \tau_{\mathrm{hl}})$ drops significantly after training on $\mathcal{D}(X_{\mathrm{tr}}^{\mathrm{adv}}, \tau_{\mathrm{hl}})$. This suggests the network chooses to learn the cue in $\tau_{\mathrm{d}}$, rather than $\tau_{\mathrm{hl}}$, as it predicts $1 - \tau_{\mathrm{hl}}$ on $X_{\mathrm{te}}^{\mathrm{adv}}$, which coincides with $\tau_{\mathrm{d}}$. We note that we keep the class balance and sizes of the random and adversarial splits approximately the same (see Fig. 5-center).

A discovered task $\tau_{\mathrm{d}}$, in this case, can be seen as a spurious feature [68, 33] and the adversarial split creates a spurious correlation between $\tau_{\mathrm{hl}}$ and $\tau_{\mathrm{d}}$ on $X_{\mathrm{tr}}^{\mathrm{adv}}$, that fools the network. Similar behaviour was observed before on datasets where spurious correlations were curated manually [67, 47, 77]. In contrast, the described approach using the discovered tasks allows us to find such spurious features, to which networks are vulnerable, *automatically*. It can potentially find spurious correlations on datasets where none was known to exist or find new ones on existing benchmarks, as shown below.

In Appendix F, we show how to extend this approach to multi-class classification, and provide results for the results for the original 10-way classification task of the CIFAR-10 dataset. We also construct and present adversarial adversarial splits for ImageNet and CelebA datasets in Appendix G.

### 6.2 Neural Networks Favor Learning the Task with a Higher AS.

The empirical observation made in the previous section demonstrates that a network trained on a dataset where $\tau_{\mathrm{hl}}$ and $\tau_{\mathrm{d}}$ coincide predicts the latter. While theoretical analysis is needed to understand the cause of this phenomenon, here, we provide an empirical clue towards its understanding.

We consider a set of 10 discovered tasks and 4 human-labelled tasks. For all pairs of tasks $(\tau_1, \tau_2)$ from this set, we construct the adversarial split according to Eq. 6, train a network on $\mathcal{D}(X_{\mathrm{tr}}^{\mathrm{adv}}, \tau_1)$ and test it on $\mathcal{D}(X_{\mathrm{te}}^{\mathrm{adv}}, \tau_1)$. Fig. 5-right shows the test accuracy against the difference in the agreement

score of these two tasks $AS(\tau_1) - AS(\tau_2)$. We find that the test accuracy w.r.t. $\tau_1$ correlates well with the difference in the agreement: when $AS(\tau_1)$ is sufficiently larger than $AS(\tau_2)$, the network makes predictions according to $\tau_1$, and vice-versa. When the agreement scores are similar, the network makes test predictions according to neither of them (see Appendix I for further discussion). This observation suggests that an adversarial split is successful, i.e., causes the network to fail, if the AS of the target task $\tau_1$ is lower than that of the task $\tau_2$ used to create the split.

# 7   Conclusion

In this work, we introduce *task discovery*, a framework that finds tasks on which a given network architecture generalizes well. It uses the Agreement Score (AS) as the measure of generalization and optimizes it over the space of tasks to find generalizable ones. We show the effectiveness of this approach and demonstrate multiple examples of such generalizable tasks. We find that these tasks are not limited to human-labelled ones and can be based on other patterns in the data. This framework provides an empirical tool to analyze neural networks through the lens of the tasks they generalize on and can potentially help us better understand deep learning. Below we outline a few research directions that can benefit from the proposed task discovery framework.

**Understanding neural networks' inductive biases.** Discovered tasks can be seen as a reflection of the inductive biases of a learning algorithm (network architectures, optimization with SGD, etc.), i.e., a set of preferences that allow them to generalize on these tasks. Therefore, having access to these biases in the form of concrete tasks could help us understand them better and guide the development of deep learning frameworks.

**Understanding data.** As discussed and shown, task discovery depends on, not only the learning model, but also the data in hand. Through the analysis of discovered tasks, one can, for example, find patterns in data that interact with a model's inductive biases and affect its performance, thus use the insights to guide dataset collection.

**Generalization under distribution shifts.** The AS turned out to be predictive of the cues/tasks a network favors in learning. The consequent adversarial splits (Sec. 6) provide a tool for studying the biases and generalization of neural networks under distribution shifts. They can be constructed automatically for datasets where no adverse distribution shifts are known and help us to build more broad-scale benchmarks and more robust models.

# 8   Limitations

The proposed instantiation of a more general task discovery framework has several limitations, which we outline below, along with potential approaches to address them.

**Completeness:** the set of discovered tasks does not necessarily include *all* of the tasks with a high AS. Novel optimization methods that better traverse different optima of the optimization objective Eq. 3, e.g., [61, 50], and further scaling are needed to address this aspect. Also, while the proposed encoder-based parametrization yields an efficient task discovery method, it imposes some constraints on the discovered tasks, as there might not exist an encoder such that the corresponding set of tasks $T_{\theta_e}$ contains *all and only* high-AS ones.

**Scalability:** the task discovery framework relies on an expensive meta-optimization, which limits its applicability to large-scale datasets. This problem can be potentially addressed in future works with recent advances in efficient meta-optimization methods [49, 65] as well as employing effective approximations of the current processes.

**Interpretability:** As we discussed and demonstrated, analysis of discovered high-AS tasks can shed more light on how the neural networks work. However, it is the expected behavior that not all of these tasks may be easily interpretable or "useful" in the conventional sense (e.g. as an unsupervised pre-training task). This is more of a consequence of the learning model under discovery, rather than the task discovery framework itself. Having discovered tasks that exhibit such properties requires additional inductive biases to be incorporated in the learning model. This was discussed in Sec. 5.2.

**Acknowledgment:**   We thank Alexander Sax, Pang Wei Koh, Roman Bachmann, David Mizrahi and Oğuzhan Fatih Kar for the feedback on earlier drafts of this manuscript. We also thank Onur Beker and Polina Kirichenko for the helpful discussions.

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
