# OpenReview forum: "Task Discovery: Finding the Tasks that Neural Networks Generalize on"
_NeurIPS.cc/2022/Conference — NeurIPS 2022 Accept_

### Official Review · Reviewer_G4XU · 2022-06-15

**Rating:** 7
**Confidence:** 3
**Soundness:** 3 good
**Presentation:** 3 good
**Contribution:** 3 good

**Summary:**

This paper introduces task discovery, which aims to find tasks that the model can generalize on.


**Questions:**

see Weaknesses

**Limitations:**

see Weaknesses

**Strengths And Weaknesses:**

**Strengths**

1. The task is interesting. Compared to existing works that mostly focus on optimizing models to achieve SOTA performance on several benchmarks, this task is more interesting and can help us understand NN more deeply.

2. I think the construction of  ``adversarial train-test splits’’ is valuable for understanding current out-of-distribution generalization or domain generalization problems.

3. The experiments are pretty good and basically answer most of my questions about the proposed task.

**Weaknesses**

1. Just as the author mentioned, one of my main concerns of me is the expensive meta-optimization.
2. I think the proposed task discovery task is relevant to domain generalization without demographics [1], optimal experimental design [2], and data-centric machine learning. Maybe more discussion should be included.

[1] Environment Inference for Invariant Learning, ICML, 2021

[2] Deep Adaptive Design: Amortizing Sequential Bayesian Experimental Design, ICML, 2021

---

> ### Author Response · Authors · 2022-08-02
> **Response to Reviewer G4XU**
>
> Thank you for your time reviewing our paper. We discuss the main points you raised below.
>
> ## Adversarial Splits
> We are glad that you found the construction of adversarial splits based on the discovered tasks valuable. We would like to draw your attention to the extension of the adversarial splits from the main paper to multi-class classification target tasks described in Appendix F. We also include additional results on adversarial splits for ImageNet with the visualization of the split in Fig. 7 in Appendix F. We summarize the quantitative results in the table below.
>
>
> | Split       | CIFAR-10 10-way | ImageNet 1000-way |
> |-------------|:---------------:|:-----------------:|
> | Random      |       0.78      |        0.59       |
> | Adversarial |       0.41      |        0.29       |
>
>
> ## Expensive meta-optimization
> > Just as the author mentioned, one of my main concerns of me is the expensive meta-optimization.
>
> We agree that meta-optimization techniques are generally more computationally and memory expensive than standard optimization methods. In this work, we applied the following techniques to reduce its costs:
> 1. *shared embedding space design* decreases memory costs and allows to optimize for multiple tasks at once, amortizing optimization costs among them and making it more efficient than discovering separate tasks sequentially (see also the extended discussion in the answer for Q1 for the reviewer YjqC).
> 2. *gradient checkpointing*  significantly reduces the memory demand in the inner loop.
>
> These techniques combined allow us to further extend this framework to perform task discovery on the TinyImageNet, which we include in Appendix J in the revision. We also believe that more efficient meta-optimization techniques (e.g., [3]) can be utilized in future work to scale the proposed framework even further.
>
> [3] Optimizing millions of hyperparameters by implicit differentiation, AISTATS 2020
>
> ## Additional discussion on related works
>
> > I think the proposed task discovery task is relevant to domain generalization without demographics [1], optimal experimental design [2], and data-centric machine learning. Maybe more discussion should be included.
>
> Thanks for drawing our attention to these works. We will include additional discussion using the additional space if the paper is accepted.
>
> While the main goals of our work and [1] are different as we aim to find generalizable tasks to better understand the interaction between models and data, the EI stage of the proposed EIIL algorithm is relevant to the proposed construction of adversarial splits based on the discovered tasks. We would like to highlight the following important differences in our approaches.
> 1. *Prior knowledge needed.* The EI stage of the EIIL approach proposed in [1] relies on whether the reference model $\Phi$ relies on spurious features or not (which is also discussed in Sec. 3.2 in [1]). Task discovery, on the other hand, makes a weaker assumption on the required prior knowledge and needs only to know the network architecture being used and finds features (tasks in our terminology) that can be used as spurious ones for this architecture.
> 2. *Diversity.* One of the key aspects of task discovery is finding different dissimilar tasks, which, in this context, implies finding different spurious features, while EIIL finds only one.

---

### Official Review · Reviewer_YjqC · 2022-07-11

**Rating:** 8
**Confidence:** 4
**Soundness:** 3 good
**Presentation:** 4 excellent
**Contribution:** 4 excellent

**Summary:**

This paper proposes a new paradigm in which given a fixed model architecture (with learnable parameters and different initializations) the goal is to automatically discover a "task", or rather a binary label for inputs in a given dataset. This is achieved by learning a task network via optimizing a proposed metric termed the agreement score. This metric assesses on average how often two different models of the same class trained on the same task will agree with one another on unseen data, with the premise being that if the class of models generalize well then they should be in agreement more often on unseen data. This metric is proposed for both automatically learning tasks as mentioned before as well as being a investigatory tool for analyzing different biases and settings.

**Questions:**

I was wondering if the authors could speak more to their decisions regarding the approach taken for learning multiple tasks simultaneously. In your findings, was having a shared encoder with task-specific linear discriminators sufficiently flexible? Did you happen to notice the embeddings to be particularly sparse in high-dimensions? I am curious if a more flexible task-specific operation was considered, such as a network that accepts the shared encoding and a learned task-specific embedding or even just learning a task-specific shallow network that accepts the shared encoding?

Finally, just to confirm my own intuition, when discovering a new task an entire dataset is essentially being partitioned into two classes ($t_\theta(x)\approx 1$ and $t_\theta(x)\approx 0$). In your findings, was is common that this process had a preference for discovering tasks with high rates of separability or decisiveness (meaning for a random $x\sim X$ it is highly likely for $t_\theta(x)$ to be close to 0 or 1) or did it also allow for instances where there are larger subsets of the data where the class is unsure ($t_\theta(x)\approx 0.5$)? Additionally, would you say that this process tended to prefer discovering tasks with roughly equal class balance? It would be interesting to be able to control or target the degrees of class imbalance exhibited within a class as I could see that leading to uncovering some uncommon but distinct traits or features within a dataset.

**Limitations:**

The authors have adequately discussed the limitations of their approach and findings.

**Strengths And Weaknesses:**

The proposal is novel and interesting as both a tool for analysis as well as for automatic task discovery. The paper has a good mix of technical contributions, ablation studies / investigations, as well as qualitative demonstrations. The methodology developed is well supported and motivated. I found the premise, proposed techniques, and experiments to all be clear and concise.

One slight correction should be made to the equations throughout the paper. Often within an expected value there is something like $\tau_1(x)=\tau_2(x)$ such as Eq. 2. These all need to be inputs to an indicator function to be proper.

---

> ### Author Response · Authors · 2022-08-02
> **Response to the reviewer YjqC (1/2)**
>
> Thank you for your time reviewing our paper. Below, we answer your questions.
>
> ## Q1: The choice of the shared embedding space for discovering multiple tasks
> > I was wondering if the authors could speak more to their decisions regarding the approach taken for learning multiple tasks simultaneously. In your findings, was having a shared encoder with task-specific linear discriminators sufficiently flexible? [...] I am curious if a more flexible task-specific operation was considered, such as a network that accepts the shared encoding and a learned task-specific embedding or even just learning a task-specific shallow network that accepts the shared encoding?
>
> Thanks for the interesting question! The response below will hopefully make our design choice and its comparison to other possibilities, including those mentioned by you, clearer.
>
> * In general, the shared embedding formulation should be flexible enough given high enough dimensionality. Indeed, if $d \geq N - 1$, then any shuttering of $N$ embeddings is possible with a linear classifier, i.e., any set of tasks can be represented with this architecture.
> * In practice, of course, we use much lower dimensionality and a specific encoder architecture to extract embeddings, which might introduce some constraints. In our experiments, however, we did not find it less flexible than modeling each task with a separate task-network while having additional benefits, as discussed below.
>
>
> In the first version of task discovery that we attempted initially, we modeled every task with a separate task-network and discovered tasks sequentially, adding new ones to the pool. For each new task, we optimized its AS while ensuring it is dissimilar to previously found tasks in the pool via a similarity loss.
>
> This approach is very flexible as it models each task with an independent network. However, the shared embedding space with linear heads has the following benefits compared to its more flexible counterparts.
> * *Analytical dissimilarity between tasks*:
>     * In order to control the similarity of tasks modeled with separate networks, we need to obtain predictions for each object for each task and force them to be dissimilar. This increases computational costs quadratically with the number of tasks.
>     * In the case of shared embedding space, we are able to control similarity between tasks by inducing uniformity over the shared embedding space. In this case, we know that two tasks corresponding to two perpendicular hyperplanes (as shown in Fig. 3-right) will be dissimilar.
>     * Note that in the case of more complex task-specific heads, e.g., shallow networks, we (to the best of our knowledge) would not be able to control the similarity analytically even if the uniformity was induced. Therefore, the time and compute would still scale quadratically.
> *  *Discovering a space of tasks*:
>     *  There are potentially too many tasks with a high AS out of all tasks to list all of them. It is, therefore, beneficial to approach this problem differently and, instead, construct a "**basis**" set of tasks and an operation to combine those tasks that keeps the AS high, which together span a *space* of high-AS tasks. This leads to a massive boost in efficiency in terms of the total number of discovered high-AS tasks vs. the spent compute.
>     *  The shared embedding space with linear heads can be seen as a step towards creating such a space of tasks, where the basis is any orthogonal basis in the $\mathbb{R}^d$ embedding space, and the operation is the standard linear combination of vectors from $\mathbb{R}^d$.
>     *  On the other hand, when modeling each task with a separate network, we discover only a relatively small and finite set of tasks, which additionally increases the computational and memory costs per task.
> * *Amortization of the optimization's computational costs:*
>     * In addition to the lower memory costs to store all the tasks and computational costs to minimize the similarity, the shared embedding space formulation benefits from training multiple tasks simultaneously and amortizing the optimization costs through the shared encoder.
>     * For example, in the first version with independent optimization for each task (except the similarity constraint), where we find tasks sequentially, it takes around 1.5 hours to find a single task, whereas it takes only 15 hours to train the shared embedding space with $d=32$ (both on a single A100 GPU). Note that the embedding version also creates a whole space of tasks, as discussed above.
>
>
> While we found the design with the shared embedding space and linear heads both computationally efficient and flexible enough to find many high-AS tasks, it would be, indeed, interesting to explore in future work other design choices that are potentially more flexible yet efficient enough.

---

> ### Author Response · Authors · 2022-08-02
> **Response to the reviewer YjqC (2/2)**
>
> >Did you happen to notice the embeddings to be particularly sparse in high-dimensions?
>
> We did not find embeddings to be sparse, and the distribution of values at each dimension is close to a zero-mean normal distribution. Note that this is the expected behavior due to the uniformity loss (Eq.(5)) we optimize, which ensures that objects' embeddings are distributed uniformly on the unit hypersphere in the embedding space.
>
> ## Q2: On task balance and separability
>
> >Finally, just to confirm my own intuition, when discovering a new task an entire dataset is essentially being partitioned into two classes (tθ(x)≈1 and tθ(x)≈0). In your findings, was is common that this process had a preference for discovering tasks with high rates of separability or decisiveness (meaning for a random x∼X it is highly likely for tθ(x) to be close to 0 or 1) or did it also allow for instances where there are larger subsets of the data where the class is unsure (tθ(x)≈0.5)? Additionally, would you say that this process tended to prefer discovering tasks with roughly equal class balance? It would be interesting to be able to control or target the degrees of class imbalance exhibited within a class as I could see that leading to uncovering some uncommon but distinct traits or features within a dataset.
>
> * *Separability.* In our experiments, we specifically promote separability to avoid a trivial solution to the task discovery problem of assigning 0.5 probability to all objects. Therefore, most of the objects have a probability close to either 0 or 1. In order to build more intuition on the underlying separability and how it changes over the course of tasks' meta-optimization, we include videos representing this dynamic for unregulated and regulated task discoveries in the revised supplementary material (`task-discovery-viz` folder). Note that we only visualize a few images, and the dynamic of the rest might be different.
> * *Task balance.* In this work, we mainly focus on discovering balanced tasks. Specifically, the uniformity constraint along with sampling tasks-specific hyperplanes that pass through origin ensure discovering balanced tasks. However, one can easily target the desired class balance in the current framework by using hyperplanes that split the hypersphere of objects in the corresponding ratio. We provide an illustration that captures this idea in **`balance.pdf` in supplementary material**. Further, a similar idea can be used to discover multi-class tasks as we do in Appendix K.

---

> > ### Comment · Reviewer_YjqC · 2022-08-09
> > **Response**
> >
> > Thank you for the in depth replies. I believe that this area of research is very interesting and am definitely interested in seeing where it goes in the future.
> >
> > I maintain my original scores.

---

### Official Review · Reviewer_RNrr · 2022-07-12

**Rating:** 6
**Confidence:** 3
**Soundness:** 4 excellent
**Presentation:** 3 good
**Contribution:** 3 good

**Summary:**

The authors formulate an agreement score that scores how well two networks from different initializations, etc converge to a common solution on a task. They claim that the agreement score is a proxy measure for how generalizable a task is. They use this agreement score to find tasks that networks can generalize on.

**Questions:**

- How exactly is the AS being used to discover adversarial splits?
- Can the authors provide some qualitative examples for the tasks being discovered? Are they useful tasks?

**Limitations:**

The authors did include a paragraph on limitations.

**Strengths And Weaknesses:**

Strengths:
- The problem formulation is unique and probably interesting. The agreement score is also interesting since it shows how robust a network is on a task.

Concerns/weaknesses:
- I am mostly concerned about the practicality of such a problem definition. Why would one want to fix a network and find a task? Tasks are defined by practical demand and we find a solution to them, not the other way around. I believe this would be interesting if we were to find the set of tasks a network is good at, but that is rarely the use case. Searching in the opposite way makes more practical sense - which networks are good for my task?
- I am not convinced why the agreement score is a proxy measure for generalizability. It seems like it is a proxy measure for robustness of a network for a task. If a network can converge a good solution from different initializations, it means that the task space is well defined and convex and easy for the network. Doesn't necessarily mean generalization - which in my knowledge is defined as performance on unseen and novel combinations of data.
- The authors say that the agreement score based search doesn't discover any real tasks. This further convinces me that the searching in the task space is probably unnecessary.
- Discovering the adversarial splits seems like an interesting direction to me here, but I didn't quite understand how the AS is being used here. To me, it seems like we can do this with any dataset and split it in a way where test labels are starkly different or opposite to train labels for a certain feature or feature combinations.

---

> ### Author Response · Authors · 2022-08-02
> **Response to Reviewer RNrr**
>
>  We thank you for the useful feedback. Below we address the raised questions and concerns. As echoed by the other reviewers, we think the introduced task discovery problem accompanied with the experimental study contributes to the field of understanding deep learning generalization, and the proposed experimental framework of discovering generalizable tasks can serve as the backbone for many more insightful works, thus, outlining a fruitful research direction.
>
>
> ## Practicality of the task discovery problem definition
> >I am mostly concerned about the practicality of such a problem definition. Why would one want to fix a network and find a task? Tasks are defined by practical demand and we find a solution to them, not the other way around.
>
> We believe that the usefulness of a work/method for analyzing tasks is not limited to being directly used for learning a solution for a specific pre-defined task. The proposed framework is practical and useful via contributing to a better understanding of a model and data at hand and deep learning generalization in general. We refer to [1] as an example where a network was simply trained to fit labels sampled randomly (a random-labelled task). While it seemed unusual or impractical at first, it was a thought-provoking idea that led to a large number of theoretical and empirical works that deepen our understanding of deep learning [2], which justifies the usefulness and practicality of the experimental framework proposed in that paper. This paper and the extensive studies provided in it intend to contribute in this direction, as also echoed by the other reviewers.
>
>
> Besides, the introduced task discovery framework and tasks found in this work already provide and outline practical use-cases through adversarial splits, application to transfer learning (discussed with the reviewer QpNg in Q4), and understanding of the interaction between data and a network.
>
> Finally, we believe that the use of a task (labelling of a dataset in this context) is not limited to training a model to fit it and make the corresponding predictions. The use of discovered task for constructing adversarial splits proposed in this work and random-labelled tasks in [1, 2] provide such examples.
>
>
> >The authors say that the agreement score based search doesn't discover any real tasks. This further convinces me that the searching in the task space is probably unnecessary.
>
> Discovering real human-labelled tasks is not the only reason for searching in the space of tasks (especially since we already have access to them) and is not the goal of the proposed task discovery framework. In fact, we would **not** want the discovered tasks to be the human-labelled ones unless that is what the inductive bias of the network causes. Instead, it allows us to reveal *the* inductive biases of a network, not what we might want it to learn, and see what patterns in data it can learn sufficiently well, which clearly is not limited to human-labelled tasks. This is extensively discussed in Appendix E. We refer to Appendix E for the discussion on whether one should expect discovered tasks to be existing human-labelled ones or human-interpretable in general.
>
>
> Finding tasks that are *different* from human-labelled ones is especially important in other contexts, e.g., in adversarial splits (see also the answer on the connection between the AS and adversarial splits below), as we would not be able to construct one otherwise. Indeed, if a target human-labelled task and a discovered task are very similar, i.e., label objects in the same way, then almost all objects will fall into the train set, and there will not be test objects which two tasks label differently, hence, no adversarial train-test split.
>
> [1] *Understanding deep learning requires rethinking generalization*, ICLR 2017
> [2] *Understanding deep learning (still) requires rethinking generalization*, Communications of the ACM 64.3 (2021)

---

> > ### Author Response · Authors · 2022-08-02
> > **Response to Reviewer RNrr**
> >
> > ## The relation between the agreement score and generalization
> >
> > >I am not convinced why the agreement score is a proxy measure for generalizability. It seems like it is a proxy measure for robustness of a network for a task. If a network can converge a good solution from different initializations, it means that the task space is well defined and convex and easy for the network. Doesn't necessarily mean generalization - which in my knowledge is defined as performance on unseen and novel combinations of data.
> >
> > In the case of searching in the space of tasks, the connection between the agreement score and generalization, as measured by the error on hold-out test data, is governed by the bias-variance decomposition as discussed in Sec. 3.1. We also refer to the Proposition 1 in Appendix A where this connection is shown rigorously. Below, we summarize the intuition behind the proof.
> >
> >
> > A task $\tau$ defines binary labeling of train and test objects. Train and test labels affect generalization, as measured by the test error, differently. The test error can be decomposed into *bias* and *variance*. For a task to have low test error, it needs both bias and variance to be low, which can be controlled separately by train and test labels.
> > * *Controlling variance with train labels.* Note that variance does not depend on test labels but only images as it measures how consistent predictions of different models are when trained on the same train image-label pairs. Therefore, one can control this term solely by choice of train labels. This is what is done in task discovery by optimizing the AS, which is a measure of variance.
> > * *Controlling bias with test labels.* When train labels are fixed, the bias can be easily minimized to zero by changing *only* test labels (it is not the case when the task is given, but in task discovery, *we* are changing the labels). Concretely, we need to set them to the average prediction of models trained on the fixed image-label pairs.
> >
> > Therefore, to find a task, that is, the labeling of train and test sets, which is generalizable, it is sufficient to follow the following procedure:
> > 1. Maximize the AS by choosing train labels, which results in low variance. This is the role of $\tau$ in Proposition 1, and the maximization of $\text{AS}(\tau)$ is done by the task discovery framework.
> > 2. Minimize the bias by setting test labels to models' mean predictions on test images. This is the role of $\hat{\tau}$ in Proposition 1, which extends labels to the test set.
> > 3. Train and test labels together define a task $\hat{\tau}$, which has a low test error. More precisely, Proposition 1 shows that the accuracy is *at least* the AS of $\hat{\tau}$ (Eq. 3 in Appendix A).
> >
> > Thus, a high agreement score gives rise to high test accuracy in this setting, which makes the AS a proxy measure for generalizability.
> >
> >
> > ## Q1: Agreement score and adversarial splits
> >
> > >Discovering the adversarial splits seems like an interesting direction to me here, but I didn't quite understand how the AS is being used here.
> >
> > We appreciate that you find the direction of creating adversarial splits interesting, and provide a discussion below to make it more clear how they are constructed and the relation to the AS.
> >
> > > To me, it seems like we can do this with any dataset and split it in a way where test labels are starkly different or opposite to train labels for a certain feature or feature combinations.
> >
> > First, we would like to note that we *do not change the labels* of objects when constructing an adversarial split, and test labels are neither different nor opposite to train labels. We only change the *arrangement of objects* between train and test sets. For example, if an image is labelled as a "cat" in the original dataset, it will have the same "cat" label regardless of being in the train or test set. Please see visual examples of such adversarial splits in Fig. 5-center for a binary task on CIFAR-10 and Fig. 6-left for CelebA in the main paper, and in Fig. 6 for the original CIFAR-10 task and Fig. 7 for ImageNet in the updated Appendix F. This process is fundamentally different from collecting images or changing labels in order to break a network via using starkly different images/labels.

---

> > > ### Author Response · Authors · 2022-08-02
> > > **Response to Reviewer RNrr**
> > >
> > > >How exactly is the AS being used to discover adversarial splits?
> > >
> > > As described in Sec. 5.3, to construct an adversarial train-test split for a target task $\tau$ (e.g., a real human-labelled task), we use a discovered task $\tau_d$ in the way governed by Eq. 6 and visualised in Fig. 5-center. Tasks discovered for a given network architecture have high agreement scores as ensured by the task discovery optimization objective in Eq. 3. This is how the AS is related to the construction of adversarial splits.
> > >
> > > Fig. 5-left shows that tasks with high AS found by task discovery, indeed, give rise to adversarial train-test splits, i.e., the test accuracy drops significantly compared to random train-test splits.
> > >
> > > We do not know the exact mechanism why high-AS tasks lead to adversarial splits. However, we hypothesise that patterns corresponding to high-AS tasks are "easier" to learn by the network, and the discovered task used to construct an adversarial split can be seen as a spurious feature that is learned by the network instead of the target task. This is evidenced by the experimental results in Fig. 5-right: the higher the AS of the discovered task used to construct the adversarial split is, the more adversarial it is, i.e., the more significant the accuracy drop is.
> > >
> > >
> > > ## Q2: Qualitative examples of the discovered tasks
> > > > Can the authors provide some qualitative examples for the tasks being discovered?
> > >
> > > We provide examples of the discovered tasks, that is, exemplar images from both classes for each task, in Fig. 4-right of the main paper. For more examples of the discovered tasks, we kindly refer you to Fig. 2 in the Appendix and the `supmat_taskviz` folder in the supplementary material for more visuals.
> > >
> > > > Are they useful tasks?
> > >
> > > Please see our answer in "*W1: Practicality of the task discovery problem definition*" for a discussion on the usefulness of the proposed framework and discovered tasks and the answer to "*Q1: Agreement score and adversarial splits*" for the usefulness of the discovered tasks for creating adversarial splits, as well as Appendix E.

---

### Official Review · Reviewer_QpNg · 2022-07-14

**Rating:** 7
**Confidence:** 4
**Soundness:** 3 good
**Presentation:** 3 good
**Contribution:** 3 good

**Summary:**

This paper studies the problem of discovering the generalizable tasks over the models.
In this context, the image label assignment of labels is considered a different task.
Then, the method proposed in the paper seeks to find different assignments of label sets over images such that desired generalizability is ensured. In a way, the labels are merged to create new labels to simulate new tasks. Such fusion may or may not be semantically meaningful. In this work, the authors wonder if a meaningful fusion (thus the derivation of a new task) can be performed by seeking the generalizability of the derived task over the test set. The authors propose a framework for task discovery that automatically finds generalizable tasks using the proposed agreement score. Furthermore, a case of “adversarial train-test splits' ' where a split of datapoints between training and testing leads to the failure of the model is also studied.



**Questions:**

* Trivial solution: to avoid the trivial solution the proposed formulation of (3) makes a trade off between tasks and agreement. This trade-off makes it unclear what is indeed being searched, as the hypaerparameter \lambda plays the role. Obviously, the presence of hypaerparameter in itself is not a problem. In this context of this paper however, the discovery of the generalizable tasks which is unknown, becomes the function of the hyperparameter that needs to be tuned. I wonder how this hypaerparameter can be selected meaningfully as there is no ground truth (or any interpretable measure as far as I can see).

* Regulated discovery: In line 236, the figure Fig.5.2 (bottom right) is incorrectly referred (also in line 225). The regularization of simCLR is as a matter of fact simply a better visual extractor.  Such pre-training learns the some desired invariances thereby leading to similar invariances in the discovered tasks. Such regularization however needs to be task dependent. For example, if one wishes to perform rotation estimation, features learned for rotation invariance are not desired. This particular case is justified for the image classification by object class (in the used case here). However, a broader discussion on what kind of regulated discovery is meaningful needs to be provided. Here again, this question arises as one is unaware beforehand what task is being searched, which is instead searched merely based on some objective which is effectively only vaguely interpretable.

* Embedding space: Monte-Carlo gradient estimator technique resulting uniformity in dissimilar tasks, and rest of the discussion is unclear. This may either require references or more details. Is the presented claim coupled with the choice of embedding space for the task?

* Training size in the context of utility: discovery of the generalizable tasks can potentially be used for knowledge transfer across tasks. This however is meaningful in practice when only a limited amount of data from the target task is available. The proposed method appears to be heavily biased from the pretraining in the case of limited data.



**Limitations:**

There is no obvious severe limitation of this paper besides what the authors have already mentioned in the paper. I can say nothing regarding the negative social impact of this paper from my reading.

**Strengths And Weaknesses:**


Strengths:

* This paper studied an interesting problem to uncover the nature of the deep learning models by means of agreement in the test set, in the context of the multi-task learning setup for task discovery for generalization.

* The proposed hypothesis and developed framework are intuitive and convincing.

* The paper is well written and easy to follow. The unconventional writing of the abstract and the discussion-rich paper make it an interesting read.

* Experimental results, both quantitative and qualitative, validate the significance of the proposed method.

* The studied case of adversarial train-test splits for the setup of the proposed setup is particularly interesting (although not very surprising).

* Experimental results in Fig 7 in the supp. Material is meaningful. I suggest that to be included in the main paper.

* Supp. Material provided with the submission (document, code, and video) are helpful.


Weakness:

* Conclusion: The conclusion of the paper is not really helpful. It reads as few comments of discussion and largely limitation in rather a short paragraph. The conclusion does not summarize the discovery of the paper in terms of theoretical or experimental results. In fact, the conclusion provided in the supp. Video is rather more meaningful.

* Experimental setup: The experimental setup used in this paper is rather limiting. Although already highlighted in the limitation section, this becomes rather a serious concern to gauge the effectiveness of the proposed method in more practical task discovery.

* Future work: Authors' thought processes behind formulating the problem, beside understanding the neural networks, is not clear. In particular, how the proposed framework can potentially be utilized beyond the experimental setup of this paper.

---

> ### Author Response · Authors · 2022-08-02
> **Response to the reviewer QpNg (1/4)**
>
> Thank you for your time reviewing our paper. Below, we answer your questions and the points you raised.
>
> ## Q1: Avoiding trivial solutions. On the $\lambda$ hyperparameter.
>
> > Trivial solution: to avoid the trivial solution the proposed formulation of (3) makes a trade off between tasks and agreement. This trade-off makes it unclear what is indeed being searched, as the hypaerparameter \lambda plays the role. Obviously, the presence of hypaerparameter in itself is not a problem. In this context of this paper however, the discovery of the generalizable tasks which is unknown, becomes the function of the hyperparameter that needs to be tuned. I wonder how this hypaerparameter can be selected meaningfully as there is no ground truth (or any interpretable measure as far as I can see).
>
> The hyperparameter $\lambda$ in Eq.(3), indeed, spans a trade-off between how similar tasks in $T=\{t_{\theta_1}, \dots, t_{\theta_K}\}$ are and how high their average agreement score (AS) is:
> * if $\lambda=0$, one trivial solution would be to have all $K$ tasks the same with the highest possible AS.
> * when $\lambda \to \infty$, tasks from $T$ become dissimilar, and some of them will have lower AS than the maximum possible. This trade-off is not specific to the particular formulation but rather is a consequence of the limited number of high-AS tasks out of the set of all tasks.
>
> Therefore, a practitioner can look at two metrics when selecting this hyperparameter:
> 1. *Similarity* (Eq. 2) shows how different the discovered tasks are from each other. The lower the similarity is, the better the set of all high-AS tasks is covered and, therefore, can be seen as a measure of recall.
> 2. *Average agreement score* shows whether discovered tasks are from the set of all high-AS tasks (i.e., on which a network generalizes well) and can be seen as precision.
>
> Depending on whether one needs to find just a single task with the highest possible AS or find as many tasks as possible, the 1st or the 2nd metric can be preferred.
>
> In Appendix I, Fig. 10, we provide additional results to show how such a trade-off looks in practice for the proposed embedding space formulation. We run task discovery with different values of $\lambda$ and show the results in the Similarity-AS coordinates. In fact, we see that the AS is not very sensitive to this hyperparameter and stays relatively high (compared to the AS of human-labeled tasks) even when similarity approaches its minimum of 0.5.

---

> ### Author Response · Authors · 2022-08-02
> **Response to the reviewer QpNg (2/4)**
>
> ## Q2: Regulated discovery
>
> >Regulated discovery: In line 236, the figure Fig.5.2 (bottom right) is incorrectly referred (also in line 225). The regularization of simCLR is as a matter of fact simply a better visual extractor. Such pre-training learns the some desired invariances thereby leading to similar invariances in the discovered tasks. Such regularization however needs to be task dependent. For example, if one wishes to perform rotation estimation, features learned for rotation invariance are not desired. This particular case is justified for the image classification by object class (in the used case here). However, a broader discussion on what kind of regulated discovery is meaningful needs to be provided. Here again, this question arises as one is unaware beforehand what task is being searched, which is instead searched merely based on some objective which is effectively only vaguely interpretable.
>
> Thanks for pointing out the incorrect references, we will fix them.
>
> Indeed, SimCLR and other augmentation-based contrastive pre-training methods use a curated set of augmentations as the source of supervision to learn visual representations. Whether they will be useful or not depends on a particular downstream task.
>
> The same holds true for the SSL-regulated task discovery as it only provides a tool to find tasks that are invariant to particular augmentations out of all generalizable tasks. One, therefore, needs to include only those augmentations that give rise to the invariances that are deemed to be obeyed by the desired set of tasks to be discovered.
>
> For example, if out of all generalizable tasks, one wants to find only those that are not color-based, then they can use representations invariant to color augmentations (e.g., learned with SimCLR). Note that in this case, we only use these representations as inputs to the task-network to limit tasks that can be discovered, whereas the two networks for the agreement score are still trained using raw pixels as we are interested in tasks on which the network will generalize when trained on original images, not the pre-trained representations.
>
> Thanks for pointing this out, we will include the additional discussion on the regulated task discovery using the additional space if the paper is accepted.
>
> We also note that using contrastive pre-training is only one way to regulate task discovery, and one can include additional loss terms to the original formulation in Eq.(3) or architectural constraints to guide the discovery differently. For example, one can limit the architecture of the task-network or fix the labels of some objects if additional annotations are available.
>
> ## Q3: Embedding space. Sampling dissimilar tasks
>
> > Embedding space: Monte-Carlo gradient estimator technique resulting uniformity in dissimilar tasks, and rest of the discussion is unclear. This may either require references or more details.
>
> * *Monte Carlo gradient estimator.* The formulation in Eq.(3) requires us to optimize the AS averaged over the set of tasks $\mathbb{E}_{t_\theta \sim T} AS(t_\theta)$. Calculating the AS for all the tasks at every iteration is intractable due to memory and computational costs. Therefore, we sample only a single task at a time which gives us an unbiased gradient estimation of the true gradient of the exact expectation. We refer to this as the Monte Carlo gradient estimator in L195.
> * *Sampling dissimilar tasks via inducing uniformity.* What makes tasks dissimilar in this case is the uniformity constraint and the way we sample task-specific hyperplanes $\theta_l$. If the embeddings are distributed uniformly over the hypersphere (ensured by the uniformity regularizer in Eq.(5)), then two tasks corresponding to two perpendicular hyperplanes passing through the origin (see $\theta_l^1,\theta_l^2$ in Fig.3-right) will have a minimum possible similarity of 0.5, i.e., will be dissimilar. Therefore, sampling perpendicular hyperplanes along with the uniformity constraint result in dissimilar tasks.
>
> Thanks for pointing out the lack of clarity here. We will update the paper accordingly.
>
> > Is the presented claim coupled with the choice of embedding space for the task?
>
> * The Monte Carlo gradient estimation of the expected AS is generally applicable to any task discovery formulation of the form presented in Eq.(3).
> * The dissimilarity due to sampling perpendicular hyperplanes is coupled with the choice of the embedding space formulation and the uniformity constraint as $L_\text{sim}$. In fact, this also allows us to improve computational complexity: instead of imposing the dissimilarity directly on labels provided by each task, we impose it through the embedding space, which is shared across all tasks (see Q1 answer for the reviewer YjqC for more details).

---

> ### Author Response · Authors · 2022-08-02
> **Response to the reviewer QpNg (3/4)**
>
> ## Q4: Training size in the context of utility. Transfer learning
>
> >Training size in the context of utility: discovery of the generalizable tasks can potentially be used for knowledge transfer across tasks.
>
> A large set of different tasks found by task discovery can, indeed, open new avenues for the research on transfer learning, for example:
> * *A tool for analysis.* One possibility would be to use the set of discovered tasks as an extension to those of human-labeled ones to facilitate the research on understanding when does transfer learning help and developing better "transferability" metrics.
> * *Tasks for pre-training.* Another avenue is to use discovered tasks to extend the set of source tasks for pre-training and incorporate biases. For example, one can pre-train a network of one architecture, e.g., a transformer, on a task(s) discovered for another one, e.g., a convolutional network, to transfer architecture-specific inductive biases -- which may be otherwise difficult to accomplish solely by architecture re-design. In this case, however, additional guidance might be needed to choose which of the discovered task(s) would be the best for pre-training, as trying all of them could be time-consuming.
>
>
> >This however is meaningful in practice when only a limited amount of data from the target task is available. The proposed method appears to be heavily biased from the pre-training in the case of limited data.
>
> Note that the task discovery framework only requires *unlabelled images*. It is, therefore, relatively easy to increase the dataset size as no human annotations are required.
>
> In addition, it can be seen from *Fig. 2-center* of the main paper that a relatively small number of images ($10^3$ in this case) is already enough to distinguish between semantic human-labelled tasks and random-labelled ones and, hence, perform meaningful task discovery. One, therefore, can apply task discovery even when unlabelled data is relatively scarce.
>
> ## Experimental setup
> > Experimental setup: The experimental setup used in this paper is rather limiting. Although already highlighted in the limitation section, this becomes rather a serious concern to gauge the effectiveness of the proposed method in more practical task discovery.
>
> We agree that meta-optimization techniques are generally more computationally and memory expensive than standard optimization methods. In this work, we applied the following techniques to reduce its costs:
> 1. *shared embedding space design* decreases memory costs and allows us to optimize for multiple tasks simultaneously, amortizing optimization costs among them, making it more efficient than discovering separate tasks sequentially (see also the extended discussion in the answer for Q1 for the reviewer YjqC).
> 2. *gradient checkpointing*  significantly reduces the memory demand in the inner loop.
>
> These techniques combined allow us to further extend this framework to perform task discovery on the TinyImageNet, which we include in Appendix J in the revision. We also believe that more efficient meta-optimization techniques (e.g., [1]) can be utilized in future work to scale the proposed framework even further.
>
> In addition, we were able to extend the same framework to the case of 1) Multi-class task discovery (see additional results in Appendix K) and 2) constructing adversarial splits for larger datasets such as CelebA included in the main paper and ImageNet added in the rebuttal revision (Appendix F.3).
>
> [1] Optimizing millions of hyperparameters by implicit differentiation, AISTATS 2020

---

> ### Author Response · Authors · 2022-08-02
> **Response to the reviewer QpNg (4/4)**
>
> ## Conclusion and future work
> >Conclusion: The conclusion of the paper is not really helpful. It reads as few comments of discussion and largely limitation in rather a short paragraph. The conclusion does not summarize the discovery of the paper in terms of theoretical or experimental results. In fact, the conclusion provided in the supp. Video is rather more meaningful.
>
> We are glad that you found the supplementary video helpful and meaningful. We will extend the discussion on the conclusions and limitations using the additional space if the paper is accepted.
>
> >Future work: Authors' thought processes behind formulating the problem, beside understanding the neural networks, is not clear. In particular, how the proposed framework can potentially be utilized beyond the experimental setup of this paper.
>
> In this work, we mainly focused on developing the task discovery framework that can find a diverse and large set of tasks based on as few assumptions as possible and its implications for understanding the interaction between data and neural networks optimized with SGD. We believe that the proposed framework is not limited to this application only. Below, we summarize a number of its potential utilities for other fields of deep learning. In addition, we think the discussion on human-interpretability and "usefulness" of the discovered tasks provided in the **Appendix E**, which is based on the insights we gathered over the trajectory of this project, is useful for understanding what one may and may not expect out of a task discovery framework. We recommend seeing that section.
>
> * *Understanding Data.* Task discovery not only sheds more light on a network but also the data at hand. Through analyzing the discovered tasks, one can find insights about given data, e.g., what are the patterns present in data that can be learned and which of them can be spuriously correlated with a target task (e.g. the adversarial splits). These insights can further guide a data collection and interpretation.
> * *Transfer/Meta-/Few-shot/Multi-task Learning.*  Multiple machine learning fields focus on building models and algorithms that can tackle many different tasks. However, many of these works are bottlenecked by a small set of tasks defined and labeled by humans, which can limit or bias the development of more successful methods. For example, in few-shot classification, a seemingly large set of tasks is usually artificially constructed out of a relatively small set of human-labeled classes (e.g., calling each possible superset of the 1K classes in ImageNet a new "task"), which can lead to overfitting, i.e., poor generalization to new tasks [1]. Building a more diverse set of tasks not limited to human-labeled ones can help to address this problem and lead to better-performing methods. Also, as discussed in the Q4 above, having more tasks can lead to a better understanding of these methods.
> * *Empirical Understanding of Deep Learning.*  A large line of works tries to fill the gap between the success and our understanding of deep learning through analysis of empirical phenomena [2]. Task discovery can allow us to evaluate found phenomena on a broader set of tasks not limited to human annotations. For example, a double-descent phenomenon [3] is mostly studied when a network is trained on a fixed human-labeled task. An interesting question would be how this behavior changes across tasks.
>
>
> [1] Meta-Learning Requires Meta-Augmentation, NeurIPS 2020
>
> [2] Identifying and Understanding Deep Learning Phenomena, ICML 2019 workshop
>
> [3] Reconciling modern machine learning practice and the bias-variance trade-off, Proceedings of the National Academy of Sciences 116.32 (2019)

---

### Author Response · Authors · 2022-08-02
**Summary of the revision and responses**

We thank the reviewers for their time, thoughtful reviews, and suggestions that will help to further improve the final version of the manuscript. We appreciate that the reviewers found the proposed task discovery formulation interesting (YjqC, RNrr, QpNg, G4XU), novel and unique (YjqC, RNrr); the experiments validating the significance of the proposed method (QpNg), answering most questions (G4XU), and the paper being clear, concise, well written and easy to follow (YjqC, QpNg). We are also glad to see that the reviewers found the proposed adversarial splits valuable and interesting (G4XU, QpNg).

We respond to each reviewer individually to answer specific questions and clarifications. Below, we briefly summarize the new provided results, questions, and our response.

## Additional results
For the rebuttal, we added the following results to the revised supplementary material file:
* **Appendix J** -- task discovery results on TinyImageNet.
    * *Fig. 11* -- examples of discovered tasks.
* **Appendix K** -- the extension of the proposed task discovery method to the multi-class classification tasks.
    * *Fig. 12* -- examples of discovered 10-way tasks.
* **Appendix F.3** -- discovered adversarial splits for the ImageNet dataset.
    * *Fig. 6* -- examples of adversarial splits for the CIFAR-10 (10-way target task).
    * *Fig. 7* -- an example of an adversarial split for ImageNet.
* **Appendix I** -- analysis of how the hyperparameter $\lambda$ influences task discovery.
    * *Fig. 10* -- similarity and AS for tasks discovered with different values of $\lambda$.
* `task-discovery-viz` -- a directory containing videos of tasks trajectories throughout the task discovery optimization process:
    * `task-discovery-viz/unregulated` -- tasks for unregulated task discovery
    * `task-discovery-viz/ssl-regulated` -- tasks for SSL regulated task discovery

## Summary of responses
* Reviewer **QpNg** asked to provide a more in-depth discussion for regulated task discovery and future work, which we do in the answer and will include in the final version of the paper. We also provide additional experimental results on the influence of the hyperparameter $\lambda$ in response to the corresponding question.
* Reviewer **YjqC** raised the questions about modeling multiple tasks with a shared encoder and linear heads, which we answer by describing the benefits of this approach compared to more flexible counterparts and comparing it to the performance of a more flexible method used in our preliminary experiments.
* Reviewer **RNrr** raised the concern regarding the practicality of the proposed method, which we address and discuss in the response. We also provide further clarifications on the connection between the AS and generalization and the AS and adversarial splits in addition to the description in the paper.
* Reviewer **G4XU** asked to include additional discussion on the relevant work, which we will do using the additional space in the camera ready. We also discuss the expensive meta-optimization aspect.

---

### Meta-Review · Area_Chair_2ViV · 2022-08-25

**Recommendation:** Accept
**Confidence:** Certain

**Metareview:**

This paper contributes an interesting point of view: given a particular model, can we find tasks for which it will be good?  For this end, the paper proposes an agreement score whose applications include finding generalizable sub-tasks and adversarial train-test splits.  All reviewers were in favor of acceptance, generally agreeing that the paper provides theoretical and practical contributions.

**Award:**

No

---

### Decision · Program_Chairs · 2022-09-14

Accept